# Relationship between adolescent anemia and school attendance observed during a nationally representative survey in India

Jan-Walter De Neve [1] ✉, Omar Karlsson[2,3], Rajesh Kumar Rai [4,5,6,7,8], Santosh Kumar [9] & Sebastian Vollmer [6,7]

## Abstract

**Background** Anemia has been suggested to be related with schooling outcomes in India. Less is known, however, about whether the observed relationship persists after accounting for all household-level factors which may confound the association between anemia and schooling.

**Methods** Nationally representative data on adolescents aged 15–18 years with data on measured hemoglobin level and school attendance were extracted from India's National Family Health Surveys conducted between 2005 and 2021. We compared school attendance between adolescents living in the same household but with varying levels of hemoglobin concentration, while controlling for age and period effects. We assessed heterogeneity in the relationship between anemia and school attendance across anemia severity groups and socio-demographic characteristics.

**Results** The proportion of adolescents with any anemia is 55.2% (95% CI: 55.0–55.5) among young women and 31.0% (95% CI: 30.6–31.5) among young men. In conventional (between-household) regression models, having any anemia is associated with a 2.5 percentage point reduction (95% CI: 2.1–2.8) in school attendance; however, in household fixed-effects models, anemia has qualitatively small and non-significant effects on school attendance. Our results are consistent using alternative model specifications as well as across anemia severity groups, genders, types of relationship to the household head, household wealth quintiles, and states and union territories in India.

**Conclusions** This within-household analysis finds little evidence that anemia is associated with school attendance among adolescents in India. Observational studies likely overstate the connection between anemia and school attendance due to household factors that have not been accounted for.

## Plain language summary

Anemia is a condition which leads to a decreased capacity to circulate oxygen in the body resulting in fatigue, weakness, dizziness, and shortness of breath among other symptoms. It has been proposed that having anemia can impact the education of adolescents. We undertook a large-scale study of the relationship between anemia and school attendance among adolescents in India. We found that household-level factors are linked with school attendance, and the direct relationship between anemia and attendance was less clear. This study highlights the need to consider all influences that can impact whether adolescents can access education. Thus, approaches that just target adolescents at risk of anemia may not be sufficient to considerably improve school attendance at the population level in India.

---

The United Nations' Sustainable Development Goals (SDGs) call for universal primary and secondary education by the year 2030. While near universal primary school enrollment has been achieved in recent decades, substantial gaps remain at the secondary school level. In 2020, about one out of four adolescents of secondary school age was out-of-school globally[1].

These gaps in secondary school enrollment are particularly noticeable in low- and middle-income countries (LMICs)[2]. In India, for instance, over 40 million youth of upper secondary age were not in school as of 2020, the largest population of adolescents out-of-school in any country worldwide[1]. Additionally, the COVID-19 pandemic has led to large losses of income and

[1]Heidelberg Institute of Global Health, Faculty of Medicine and University Hospital, University of Heidelberg, Heidelberg, Germany. [2]Department of Economic History, Lund University, Lund, Sweden. [3]Duke University Population Research Institute, Durham, NC, USA. [4]Department of Global Health and Population, Harvard T. H. Chan School of Public Health, Harvard University, Boston, MA, USA. [5]Society for Health and Demographic Surveillance, Suri, West Bengal, India. [6]Department of Economics, University of Goettingen, Göttingen, Germany. [7]Centre for Modern Indian Studies, University of Goettingen, Göttingen, Germany. [8]Institute of Nutrition, Mahidol University, Nakhon Pathom, Thailand. [9]Keough School of Global Affairs, University of Notre Dame, Notre Dame, IN, USA. ✉e-mail: janwalter.deneve@uni-heidelberg.de

the temporary closing of schools in many LMICs[3], including India[4], making the realization of global targets for universal primary and secondary education unlikely without removing barriers to secondary school enrollment.

Micronutrient deficiencies may hinder educational attainment in LMICs. Anemia, such as iron-deficiency anemia, has been suggested as a major determinant of schooling outcomes[5,6]. Hypothesized mechanisms include lower cognitive function and developmental delays[7], decreased school attendance and participation[5], lower attention span[6], and increased infection risks[8,9]. However, despite credible epidemiological and biological links, the empirical evidence on the relationship between anemia, intelligence quotient, and schooling outcomes is mixed. A meta-analysis of five studies estimated that a 1 g/dl increase in hemoglobin in children was associated with a 1.73 increase in intelligence quotient points[10]. In contrast, substantial improvement in iron deficiency in a randomized controlled trial was not associated with improved cognitive ability of school children in India[7]. Furthermore, observational studies are vulnerable to confounding by unobserved factors that affect both anemia and schooling (such as primary household occupation and psychological traits, and other socio-cultural factors)[6,11]; whereas randomized controlled trials are limited in terms of external generalizability (e.g., because of their focus on a small geographical area) and are insufficiently powered to assess higher levels of anemia severity (such as moderate anemia or worse) which are relatively uncommon outcomes compared to mild anemia[5,7,12,13].

In this study, we use nationally representative data on over 250,000 adolescents aged 15–18 years old in India with measured hemoglobin concentration. We use multivariable linear regression models with household fixed effects to examine the association between having any anemia and current school attendance. Specifically, the household fixed effects model allows us to test whether adolescents with lower levels of hemoglobin concentration were less likely to attend school when comparing adolescents living in the same household—i.e., the same in terms of all observed and unobserved household characteristics. We also consider differences in the relationship between anemia and school attendance by adolescents' anemia severity group (including mild, moderate, severe, and life-threatening anemia), gender, type of relationship to the head of household (biological child or not), household wealth quintiles, and states and union territories in India[14]. Additionally, while we focus on school attendance as our primary outcome in this study, we assess other educational outcomes to the extent possible, including literacy, knowledge, and educational attainment. In within-household analyses, we find that the link between anemia and these schooling outcomes is less clear than previously suggested by studies which do not consider all household-level factors.

## Methods
### Data sources
Data were extracted from India's 2005–2006 National Family Health Survey (NFHS-3), 2015–2016 National Family Health Survey (NFHS-4), and 2019–2021 National Family Health Survey (NFHS-5), three cross-sectional, nationally representative household surveys with biomarker data on measured hemoglobin level and school attendance[15–17]. A total of 109,041 households were selected for the NFHS-3; a total of 601,509 households were selected for the NFHS-4; and a total of 636,699 households were selected for the NFHS-5. In the NFHS-3, all female household members aged 15–49 years and all male members aged 15–54 years were eligible to be interviewed. In the NFHS-4 and NFHS-5, all women aged 15–49 years were invited to participate while men aged 15–54 years were invited to participate in a random subsample of 15% of households. The choice to sample more women than men in the NFHS-4 and NFHS-5 was made because of the surveys' primary focus on maternal and child health. Household response rates were 98% in all three surveys (NFHS-3, NFHS-4, and NFHS-5). Individual participation rates were 95% (NFHS-3) and 97% (NFHS-4 and NFHS-5) for women, whereas individual participations rates were 87% (NFHS-3) and 92% (NFHS-4 and NFHS-5) for men. Strengths of the NFHS include high quality interviewer training, standardized data collection procedures, and consistent content over time, allowing comparability across survey years[18]. Nevertheless, we included indicators for survey year to take into account period effects and potential systematic differences in the measurement of hemoglobin and school attendance across surveys. We also ran all analyses separately for each survey year in addition to using the pooled dataset. We used unrestricted data which are publicly available upon request from the Demographic and Health Surveys (DHS) Program (https://dhsprogram.com/). Dataset requests must include a description of the proposed analysis. After reviewing our proposal, the DHS Program granted access to the data. Additional details on the surveys are presented in Supplementary Methods 1.

### Study population
India has the largest adolescent population in the world, with over 250 million adolescents aged 10–19 years in the 2011 Census[19]. We restricted our analysis to the period of late adolescence because the NFHS measured hemoglobin for individuals aged 15 and above and secondary school attendance lasts *de jure* until about age 18 years, yielding a sample with an age range of 15–18 years. Late adolescence is a critical period of development when investments in schooling can have substantial path dependency and potentially large effects across the life course[20]. We excluded pregnant adolescents and adolescents with extreme values of hemoglobin below 4 g/dl or above 20 g/dl under the assumption that these extreme values were likely due to measurement error[21]. Of 269,922 adolescents aged 15–18 years who completed the surveys, 12,452 (4.6%) had a missing hemoglobin measurement, 651 (0.2%) had missing data on current school attendance, 4164 (1.5%) reported being pregnant, and 1519 (0.6%) had extreme hemoglobin values below 4 g/dl or above 20 g/dl, leaving a final sample for analysis of 251,401 adolescents. Out of a total of 251,401 adolescents, 65,459 adolescents (26%) lived in households with more than one adolescent aged 15–18 years. Among those 65,459 adolescents, there was within-household variation in hemoglobin concentration (g/dl) for 64,037 adolescents (98%) and variation in anemia status for 36,971 adolescents (57%). Supplementary Fig. 1 shows a study participant diagram.

### Outcome measure
Our primary outcome was a binary indicator for an adolescent's school attendance at the time of the survey. We used the following questions which were a part of the household questionnaire: Did [NAME] attend school or college at any time during the 2005–2006 school year? (NFHS-3); Did [NAME] attend school or college at any time during the 2015–2016 school year? (NFHS-4); Did [NAME] attend school or college at any time during the 2019–2020 school year? (NFHS-5).

### Exposure
Our exposure was having any anemia defined as hemoglobin below 11.9 g/dl among non-pregnant women and hemoglobin below 12.9 g/dl among men. As a sensitivity analysis, we used hemoglobin (continuous in g/dl) instead of a dummy coded categorical variable for having any anemia. Additionally, following WHO's recommendations, we categorized anemia in non-pregnant women as hemoglobin between 11.0 g/dl and 11.9 g/dl (mild anemia), between 8.0 g/dl and 10.9 g/dl (moderate anemia), between 6.5 g/dl and 7.9 g/dl (severe anemia), and below 6.5 g/dl (life-threatening anemia). Anemia in men was categorized as hemoglobin between 11.0 g/dl and 12.9 g/dl (mild anemia), between 8.0 g/dl and 10.9 g/dl (moderate anemia), between 6.5 g/dl and 7.9 g/dl (severe anemia), and below 6.5 g/dl (life-threatening anemia). The NFHS measured hemoglobin concentration in all participants using a capillary blood sample from a finger prick, which was then analyzed using the portable HemoCue Hb 201+ device (HemoCue AB, Ängelholm, Sweden). Hemoglobin concentrations of all participants were adjusted for smoking status and altitude before applying these cutoffs. Smoking status was ascertained through self-report and altitude was measured separately for each primary sampling unit using global positioning satellite devices[15,16]. Additional details on the measurement of hemoglobin are available in Supplementary Methods 1.

## Control variables

For each adolescent, we extracted data on age (years), gender, the NFHS-provided household wealth index quintile (1 being the poorest, and 5 being the richest), and household location (urban vs. rural). We used a wide range of specifications for covariates in our analysis, first only controlling for a survey year indicator, then adding covariates which were determined prior to our exposure (age and gender), and finally adding more conventional covariates (household wealth and location). For our main analysis, we also controlled for all household-level factors using household fixed effects. Because our household identifiers were survey-specific, we implicitly accounted for time-varying shocks that affected adolescents in our pooled sample differently across survey years. Additionally, the use of several survey years allowed us to generate variation in year of birth for a given adolescent's age so that we implicitly also controlled for birth cohort effects in addition to age and period effects (since birth cohort = survey year minus age). In sensitivity analyses, we also considered alternative specifications of age (age squared, age cubed) to take into account potential non-linearities in the relationship with anemia severity group and school attendance. We also considered additional covariates, such as month of birth, religion, scheduled caste and tribe, and fixed effects for an adolescent's interviewer to control for potential measurement error due to systematic differences in hemoglobin concentration and school attendance across interviewers. Birth order was not collected among all adolescents in the NFHS and therefore not used in our analyses[22].

## Statistical analysis

**Relationship between anemia and school attendance.** Our analysis proceeded in five steps (Models 1–5). First, we estimated the relationship between having any anemia and school attendance controlling exclusively for survey year in ordinary least squares (OLS) regression models (a "stripped down" model) (Model 1). Second, we added potential confounders that were determined prior to our exposure, including age (years) and an indicator for gender (Model 2). Third, we added household wealth quintile and location (urban or rural) in addition to the covariates included in Model 2 (Model 3). Fourth, comparing how anemia affects school attendance across households may be confounded by a wide range of unobserved household-level actors that affect both anemia and school attendance[23]. To circumvent such biases, we measured the differential probability of attending school for adolescents with different anemia status living in the same household. To achieve this goal, we regressed an adolescents' school attendance on having any anemia and controlled for household-level characteristics by subtracting a household-level weighted mean from all independent variables, which is parametrically identical to adding indicator variables for each household (Model 4). In Model 5, we added pre-determined socio-demographic covariates (adolescent's age and gender) in addition to the household fixed effects included in Model 4. We clustered standard errors at the household level using the *xtreg* command in Stata (version 17).

## Heterogeneity by anemia severity group and socio-demographic characteristics

We determined how the relationship between anemia and school attendance varied by severity using a dummy coded variable for anemia severity group (mild, moderate, severe, and life-threatening anemia, with no anemia as an omitted reference category). We also used separate binary indicators for each anemia severity group to estimate the relationship between each severity group and school attendance using separate regression models. In addition to heterogeneity by anemia severity group, we also determined heterogeneity by gender[24], relationship to the head of household (biological child vs. children-in-law, grandchildren, niece/nephew, other) (see Supplementary Table 1 for the distribution in the types of relationship), household wealth and geographical region in India. To determine heterogeneity by region, we categorized all states and union territories of India by low, medium and high levels of state-level mean school attendance. Rajasthan, Maharashtra, and Tamil Nadu, for instance, had low (43.7–72.8%),

medium (72.9–81.8%), and high mean school attendance levels (82.8–94.2%), respectively, in the NFHS-5 (Supplementary Tables 2–4).

**Sensitivity analyses for anemia and school attendance.** In addition to the sensitivity analyses described above, we conducted several supplementary analyses to generate further confidence in our main results. First, potential misalignment between the timing of our exposure and outcome may introduce bias. We therefore looked at the subset of households where the time between the survey and dropping out of school is short. A 16-year-old adolescent, for instance, who was not in school at the time of the survey and finished 9th grade would be more informative than the same 16-year-old who finished 5th grade. Additional sensitivity analyses for the timing of our exposure and outcome are presented in Supplementary Methods 2. Second, we included respondents aged 15–24 years (as opposed to respondents aged 15–18 years) to obtain a subset of households with larger birth spacing between siblings in each household. Third, in analytical inference, the use of sample weights has been subject to some controversy[25]. We therefore added sample weights to our analysis as an additional robustness check. Fourth, we used alternative definitions of the outcome. Since adolescents with mild or moderate anemia can still attend school but may face difficulties learning, we assessed several learning outcomes which were available in the NFHS. As secondary outcomes, we therefore extracted data on literacy (binary), frequency reading, knowledge, progression through school (being age-for-grade), and educational level attained. As a proxy for knowledge acquired through, for example, the formal school curriculum or government public health campaigns, we also used an indicator for comprehensive knowledge on the transmission and prevention of tuberculosis (the deadliest infectious disease in India)[26].

## Ethical approval

Our analysis of this existing anonymized dataset in the public domain received a determination of non-human subjects research by the institutional review board of the Harvard T. H. Chan School of Public Health and so did not require a full ethical review.

## Reporting summary

Further information on research design is available in the Nature Portfolio Reporting Summary linked to this article.

## Results

### Data sample description

Table 1 shows demographic characteristics of analyzable survey participants. In the pooled sample, 51.5% (95% Confidence Interval [CI]: 51.4–51.7) had any anemia and 68.1% (95% CI: 67.9–68.3) were currently attending school. Moreover, although adolescents with severe or life-threatening anemia represented just 1.6% and 0.6% of our pooled sample, respectively, these numbers represent a large population in absolute terms in India (about 4 and 1.5 million adolescents, respectively, given a population of 250 million adolescents)[19]. Young women, in particular, were prone to adverse health and schooling outcomes, with more than half having any anemia compared to 31.3% among young men. About 11% more young women were currently not attending school compared to their male counterparts (32.4% vs. 29.2% out-of-school). The majority of adolescents in our sample had attained secondary school (85.0%). In terms of household characteristics, co-residency with other household members was common. The average household size in our sample was 6.0 listed household members. In the NFHS-5, the proportion of adolescents with any anemia ranged from 7.1% among young men in Lakshadweep union territory to 78.8% among young women in the Jammu, Kashmir and Ladakh union territories (Fig. 1). School attendance ranged from 52.2% in Gujarat state among young men to 94.4% among young women in Kerala state. Models using household fixed effects are limited to the sample of households with more than one adolescent aged 15–18 years. We therefore provide descriptive statistics for the sample of adolescents co-residing with other adolescent(s) (shown in

**Table 1 | Demographic characteristics of included participants**

| Subsample | All adolescents (n = 251,401) | All adolescents separately by gender | | Only adolescents living with > 1 adolescent (n = 65,459) |
| --- | --- | --- | --- | --- |
| | | Female (n = 213,012) | Male (n = 38,389) | |
| Anemia severity group | | | | |
| No anemia | 121,809 (48.5%) | 95,334 (44.8%) | 26,475 (69.0%) | 32,757 (50.0%) |
| Mild anemia | 64,142 (25.5%) | 54,460 (25.6%) | 9682 (25.2%) | 16,636 (25.4%) |
| Moderate anemia | 60,133 (23.9%) | 58,088 (27.3%) | 2045 (5.3%) | 14,737 (22.5%) |
| Severe anemia | 3899 (1.6%) | 3775 (1.8%) | 124 (0.3%) | 988 (1.5%) |
| Life-threatening anemia | 1418 (0.6%) | 1355 (0.6%) | 63 (0.2%) | 341 (0.5%) |
| Current school attendance | | | | |
| Yes | 171,143 (68.1%) | 143,952 (67.6%) | 27,191 (70.8%) | 43,902 (67.1%) |
| No | 80,258 (32.0%) | 69,060 (32.4%) | 11,198 (29.2%) | 21,557 (32.9%) |
| Educational level attained | | | | |
| No formal schooling | 13,358 (5.3%) | 11,850 (5.6%) | 1508 (3.9%) | 3679 (5.6%) |
| Primary school | 17,788 (7.1%) | 15,011 (7.1%) | 2777 (7.2%) | 4932 (7.5%) |
| Secondary school | 213,727 (85.0%) | 180,425 (84.7%) | 33,302 (86.8%) | 55,095 (84.2%) |
| Higher education | 6389 (2.5%) | 5607 (2.6%) | 782 (2.0) | 1719 (2.6%) |
| Data missing | 139 (0.1%) | 119 (0.1%) | 20 (0.1%) | 34 (0.1%) |
| Literacy | | | | |
| Yes | 224,567 (89.3%) | 189,636 (89.0%) | 34,931 (91.0%) | 57,610 (88.0%) |
| No | 22,554 (9.0%) | 19,848 (9.3%) | 2706 (7.0%) | 5959 (9.1%) |
| Data missing | 3834 (1.5%) | 3528 (1.7%) | 752 (2.0%) | 1890 (2.9%) |
| Age, years | | | | |
| 15 | 63,062 (25.1%) | 53,847 (25.3%) | 9215 (24.0%) | 17,147 (26.2%) |
| 16 | 62,839 (25.0%) | 53,281 (25.0%) | 9558 (24.9%) | 14,999 (22.9%) |
| 17 | 57,877 (23.0%) | 48,905 (23.0%) | 8972 (23.4%) | 14,058 (21.5%) |
| 18 | 67,623 (26.9%) | 56,979 (26.8%) | 10,644 (27.7%) | 19,255 (29.4%) |
| Number of household members | | | | |
| 1 (lives alone) | 222 (0.1%) | 130 (0.1%) | 92 (0.2%) | – |
| 2–3 | 22,449 (8.9%) | 18,144 (8.5%) | 4305 (11.2%) | 1290 (2.0%) |
| 4–5 | 102,879 (40.9%) | 85,649 (40.2%) | 17,230 (44.9%) | 20,154 (30.8%) |
| 6–7 | 76,446 (30.4%) | 66,307 (31.1%) | 10,139 (26.4%) | 24,052 (36.7%) |
| 8–9 | 30,309 (12.1%) | 26,394 (13.0%) | 3915 (10.2%) | 11,576 (17.7%) |
| 10+ members | 19,096 (7.6%) | 16,388 (7.7%) | 2708 (7.1%) | 8387 (12.8%) |
| Household wealth quintile | | | | |
| Quintile 1 (poorest) | 54,687 (21.8%) | 47,709 (22.4%) | 6978 (18.2%) | 12,816 (19.6%) |
| Quintile 2 | 59,104 (23.5%) | 50,753 (23.8%) | 8351 (21.8%) | 15,474 (23.6%) |
| Quintile 3 | 53,835 (21.4%) | 45,617 (21.4%) | 8218 (21.4%) | 14,697 (22.5%) |
| Quintile 4 | 46,226 (18.4%) | 38,497 (18.1%) | 7729 (20.1%) | 12,994 (19.9%) |
| Quintile 5 (richest) | 37,549 (15.7%) | 30,436 (14.3%) | 7113 (18.5%) | 9478 (14.5%) |
| Household location | | | | |
| Rural area | 184,96 (74.6%) | 158,806 (74.6%) | 26,161 (68.2%) | 47,613 (72.7%) |
| Urban area | 66,434 (26.4%) | 54,206 (25.5%) | 12,228 (31.9%) | 17,846 (27.3%) |

Data are unweighted n (%) using pooled data from the India NFHS-3, NFHS-4, and NFHS-5 (2005–2021).

column 5, Table 1). Out of a total of 251,401 adolescents (column 2, Table 1), 65,459 adolescents (26%) lived in households with more than one adolescent aged 15–18 years (column 5, Table 1); however, socio-demographic characteristics were generally similar across both samples.

**Relationship between having any anemia and school attendance**
In Table 2, we show OLS regressions results for the relationship between having any anemia and school attendance. Using a simplified ("stripped down") regression model, having any anemia was associated with a 4.5 percentage point reduction (95% CI: 4.2–4.9) in the probability of attending school relative to a baseline probability of attending school of 70.1% among non-anemic adolescents (Model 1, Table 2). The negative relationship between anemia and school attendance persisted when adding controls for observed socio-demographic characteristics. Adjusting for survey year, age (years), gender, household wealth, and location, having any anemia was still associated with a 2.5 percentage point reduction (95% CI: 2.1–2.8) in the probability of attending school (Model 3, Table 2). These estimates, however, may be confounded by unobserved household factors that may be

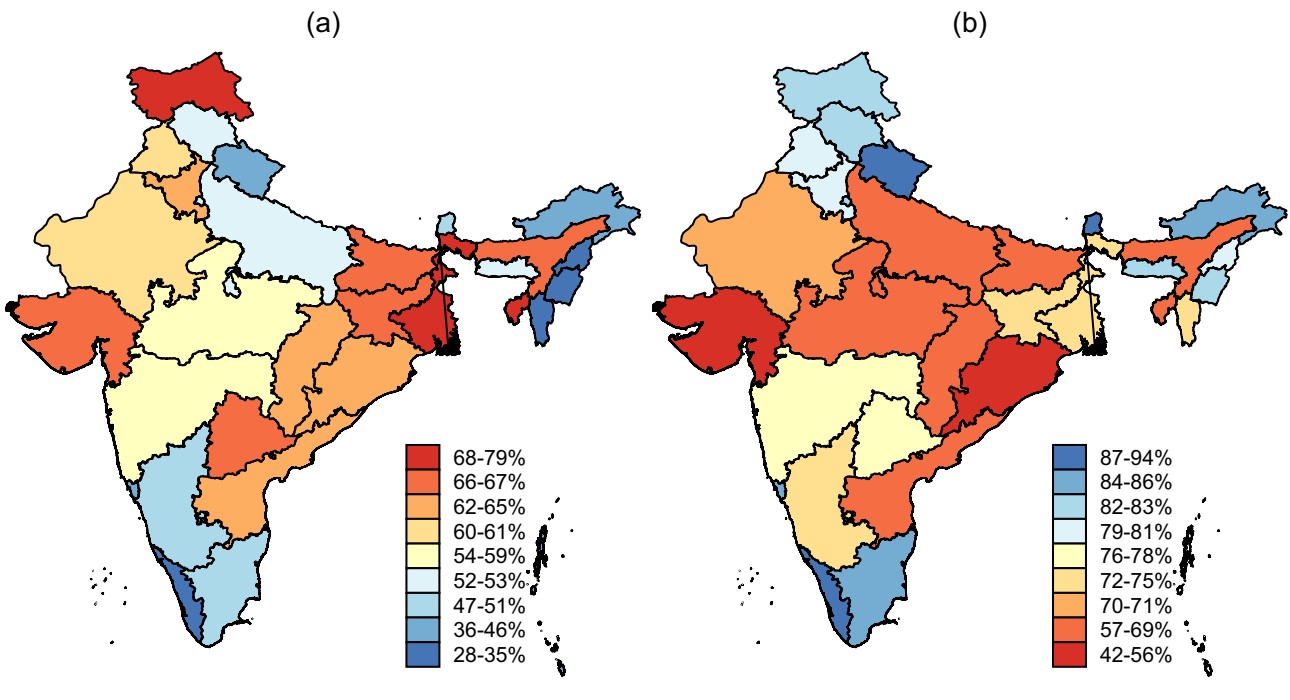

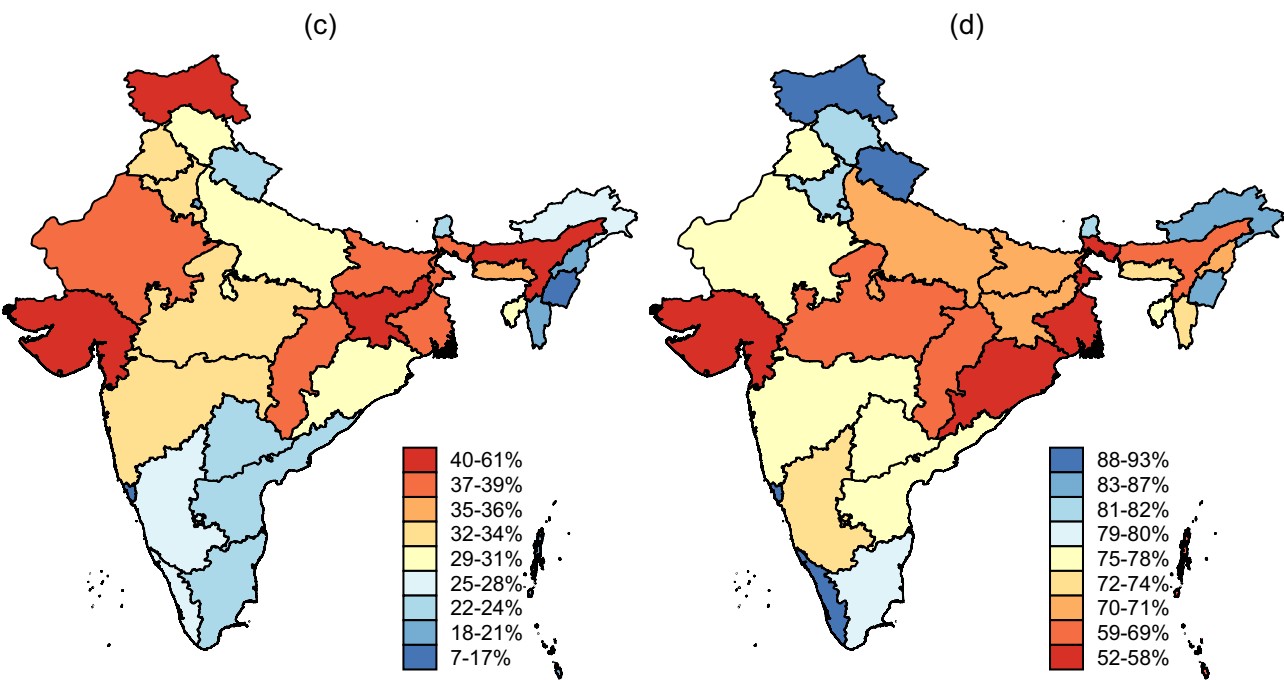

**Fig. 1 | Anemia and school attendance among adolescents aged 15–18 years by region in India.** The figure displays the proportion with any measured anemia and reported current school attendance by states and union territories among adolescents aged 15–18 years using data from the India NFHS-5 (2019–2021), separately by gender. **a** shows any measured anemia among young women, **b** shows current school attendance among young women, **c** shows any measured anemia among young men, and **d** shows current school attendance among young men. After estimating the proportion for each region, we generated groups of similar size. We then mapped anemia and school attendance for each of the groups using a base map with geographic boundaries of India provided by the Database of Global Administrative Areas (www.gadm.org). N = 107,768.

**Table 2 | Main OLS regression results for the relationship between having any anemia and current school attendance among adolescents aged 15–18 years in India, 2005–2021**

| Dependent variable (DV): Currently attending school (1 = yes, 0 = no) | (1) Indicator for survey year | (2) Adding pre-determined covariates | (3) Additional conventional covariates | (4) Household fixed effects | (5) Household fixed effects with covariates |
|---|---|---|---|---|---|
| **Predictor** | | | | | |
| Any anemia (yes = 1, no = 0) | −0.045 | −0.044 | −0.025 | 0.002 | −0.003 |
| | (0.002) | (0.002) | (0.002) | (0.004) | (0.004) |
| **Additional covariates** | | | | | |
| Survey year | ✓ | ✓ | ✓ | ✓ | ✓ |
| Age (years) | – | ✓ | ✓ | – | ✓ |
| Indicator for female | – | ✓ | ✓ | – | ✓ |
| Household wealth quintile | – | – | ✓ | – | – |
| Household location (1 = urban, 0 = rural) | – | – | ✓ | – | – |
| Household fixed effects | – | – | – | ✓ | ✓ |
| Total number of households | 217,643 | 217,643 | 217,643 | 217,643 | 217,643 |
| *N*, all households | 251,401 | 251,401 | 251,401 | 251,401 | 251,401 |
| *N*, households with >1 adolescent | 65,459 | 65,459 | 65,459 | 65,459 | 65,459 |
| Probability DV, non-anemic | 0.701 | 0.701 | 0.701 | 0.680 | 0.684 |
| *R*-squared | 0.011 | 0.065 | 0.152 | 0.001 | 0.128 |

Most prior literature has assessed the relationship between anemia and school attendance using more conventional (between-household) regression models akin to Models 1–3. The table additionally includes Models 4–5 when adding household fixed effects with and without covariates. All regressions are ordinary least squares (OLS) models. The dependent variable was a binary indicator for currently attending school at the time of the survey. Standard errors in parentheses. Sample includes survey respondents who were between 15 and 18 years old at the time of the survey and had valid hemoglobin test results. No weights were used. Source: NFHS-3, NFHS-4, and NFHS-5 (*N* = 251,401).

related to both anemia and school attendance. We, therefore, show results using an alternative empirical approach in Models 4 and 5. Regression models which take into account all observed and unobserved household characteristics suggest that the relationship between any anemia and current school attendance may be more muted than suggested by more conventional (between-household) regression models. Our estimates from household fixed-effects regression models provide little evidence (at alpha = 0.05) that having any anemia is associated with school attendance (−0.3 percentage points; 95% CI: −1.1, 0.5; Model 5, Table 2). These estimates suggest that it is unlikely that any anemia reduces the probability of being currently in school by more than 1.1 percentage points, which corresponds to the value of the lower limit of the 95% CI.

### Heterogeneity by anemia severity group and socio-demographic characteristics

Table 3 presents results for the relationship between each anemia severity group and current school attendance. Consistent with our main results, the different anemia severity groups had few detectable effects on school attendance. Having severe anemia, for example, was associated with a 5.9 percentage point reduction (95% CI: 4.5–7.4) in school attendance in more conventional (between-household) models (Model 3, Table 3); however, in household fixed effects models, having severe anemia was associated with a 0.4 percentage point reduction (95% CI: −3.4–2.5) in school attendance (Model 5, Table 3). Results were similar when using binary indicators for each anemia severity group to estimate the relationship between each severity group and school attendance using separate models (Supplementary Tables 5–7). To determine heterogeneity by gender, we also interacted each anemia severity group with a male dummy in Table 3. Coefficients for the interaction between each of the anemia severity groups and our male dummy, however, were in most cases small, suggesting that the relationship between anemia and school attendance is qualitatively similar across genders. We also show results when stratifying our sample by gender (Supplementary Tables 8 and 9), type of relationship to the household head (Supplementary Table 10), household wealth quintiles (Supplementary Table 11), and geographical region in India (Supplementary Table 12). Similar to our main results shown in Table 2, our within-household analyses

provide little evidence that anemia is associated with school attendance across all of these socio-demographic characteristics.

### Results of sensitivity analyses for anemia and school attendance

Our main results for school attendance were also consistent when using hemoglobin (continuous in g/dl) instead of a dummy coded variable for having any anemia (Supplementary Table 13); a flexible specification of age; controlling for additional covariates in our models; interviewer fixed effects; adding sampling weights; several sensitivity analyses for the timing of our exposure and our outcome (Supplementary Tables 14 and 15); and when analyzing each NFHS survey separately as opposed to using the pooled sample (Supplementary Tables 16–18). Having any anemia also had few detectable effects on our secondary outcomes. For example, having any anemia was associated with a 1.5 percentage point reduction (95% CI: 1.3–1.8) in measured literacy skills in more conventional (between-household) models; however, in household fixed effects models, having any anemia was associated with a 0.5 percentage point reduction (95% CI: −1.0–0.1) in literacy skills (Supplementary Table 19). Similarly, coefficients for the relationship between having any anemia and our other learning outcomes, such as knowledge and grade progression, were also generally qualitatively small in household fixed effects models. Taken together, these supplementary analyses further strengthen confidence in our main findings. The relationship between having any anemia and school attendance appears more muted than suggested in observational studies which do not consider all observed and unobserved household factors.

### Discussion

Anemia has been suggested as an important obstacle of human capital development in LMICs; however, most existing observational evidence does not take into account all household-level confounders[27,28], and/or is limited in terms of external generalizability[5,7,12,13]. To address this gap in the literature, we used nationally representative data with measured biomarkers for hemoglobin concentration to determine the relationship between anemia and school attendance among adolescents in India. To do so, we sought to leverage a distinct feature of the study context in our methodological approach—the fact that the average household size in India is relatively

**Table 3 | Household fixed-effects regression results for the relationship between anemia severity category and school attendance by gender among adolescents aged 15–18 years in India, 2005–2021**

| Dependent variable: Currently attending school (1 = yes, 0 = no) | (1) Indicator for survey year | (2) Adding pre-determined covariates | (3) Additional conventional covariates | (4) Household fixed effects | (5) Household fixed effects with covariates |
|---|---|---|---|---|---|
| **Predictor** | | | | | |
| No anemia | Ref. | Ref. | Ref. | Ref. | Ref. |
| Mild anemia | −0.032 | −0.032 | −0.017 | 0.008 | 0.001 |
| | (0.002) | (0.002) | (0.002) | (0.005) | (0.005) |
| Moderate anemia | −0.055 | −0.049 | −0.028 | −0.005 | −0.003 |
| | (0.002) | (0.002) | (0.002) | (0.006) | (0.006) |
| Severe anemia | −0.090 | −0.075 | −0.059 | −0.018 | −0.004 |
| | (0.008) | (0.008) | (0.007) | (0.016) | (0.015) |
| Life-threatening anemia | −0.113 | −0.095 | −0.069 | −0.051 | −0.025 |
| | (0.013) | (0.013) | (0.012) | (0.028) | (0.028) |
| **Male** | | | | | |
| No | – | Ref. | Ref. | – | Ref. |
| Yes | – | 0.056 | 0.048 | – | 0.051 |
| | – | (0.003) | (0.003) | – | (0.006) |
| **Anemia severity group*Male** | | | | | |
| No anemia*Male | – | Ref. | Ref. | – | Ref. |
| Mild anemia*Male | – | −0.022 | −0.008 | – | −0.012 |
| | – | (0.006) | (0.006) | – | (0.011) |
| Moderate anemia*Male | – | −0.010 | 0.002 | – | −0.010 |
| | – | (0.010) | (0.010) | – | (0.019) |
| Severe anemia*Male | – | 0.011 | 0.042 | – | 0.058 |
| | – | (0.040) | (0.040) | – | (0.081) |
| Life-threatening anemia*Male | – | −0.112 | −0.103 | – | -0.033 |
| | – | (0.062) | (0.060) | – | (0.070) |
| **Additional covariates** | | | | | |
| Survey year | ✓ | ✓ | ✓ | ✓ | ✓ |
| Age (years) | – | ✓ | ✓ | – | ✓ |
| Household wealth quintile | – | – | ✓ | – | – |
| Household location (1 = urban, 0 = rural) | – | – | ✓ | – | – |
| Household fixed effects | – | – | – | ✓ | ✓ |
| **Observations** | | | | | |
| Total number of households | 217,643 | 217,643 | 217,643 | 217,643 | 217,643 |
| N, all households | 251,401 | 251,401 | 251,401 | 251,401 | 251,401 |
| N, households with >1 adolescent | 65,459 | 65,459 | 65,459 | 65,459 | 65,459 |

All regressions are ordinary least squares (OLS) models. The dependent variable was a binary indicator for currently attending school at the time of the survey. Standard errors in parentheses. Sample includes survey respondents who were between 15 and 18 years old with valid hemoglobin test results. No weights were used. Source: NFHS-3, NFHS-4, and NFHS-5 (N = 251,401).
*Ref* reference category.

large. In our main analytical sample, for instance, the national median household size was six members per household. To put these numbers into context, the average household size in Germany is less than two members per household[29]. Given this context, our approach exploited variation in hemoglobin concentration among adolescents who were living within the same household and compared their school attendance as opposed to comparing adolescents who were living in different households which is the case in more conventional analyses. Our findings have several implications for the role of anemia in human capital development in LMICs.

First, the proportion of children with anemia in India has declined in the past decades[30,31]. However, our descriptive results point to a reversal of this trend among school-going aged adolescents in more recent surveys[32,33]. The proportion of adolescents aged 15–18 years with any anemia increased from 43.7% in 2005–2006 to 55.7% in 2019–2021 (Supplementary

Tables 2–4). Anemia was also a considerable problem among male adolescents in our sample. The proportion of male adolescents with any anemia was over 30%, further justifying the need to consider both young men and women in future efforts to reduce the burden of anemia among school-going age adolescents[21]. Despite the increasing burden of anemia among adolescents[33] and COVID-19 pandemic[4], the proportion of adolescents who attended school in our sample increased from 55.4% to 70.3% during the same period (Supplementary Tables 2–4).

Second, our household fixed-effects models, which filtered out all observed and unobserved household characteristics, provided little evidence that having any anemia was associated with school attendance (Table 2). Consistent with our main results, household fixed-effects models also found little evidence of a dose-response relationship between the level of anemia severity and school attendance (Table 3). Policymakers should be aware that

existing estimates from observational studies which ignore these household characteristics may be biased upwards. Reducing anemia may not substantially improve average school attendance among adolescents at the population level, and other interventions may be needed to further improve school attendance in the region. Common reasons for school absenteeism among out-of-school adolescents, for example, included low perceived benefits of additional education, financial costs associated with school, domestic activities (such as doing laundry, cooking, and cleaning at home), as well as adolescent marriage among young women (Supplementary Fig. 2). These barriers to schooling could be addressed through, for example, compulsory schooling enforcement or providing information to households about the economic and health benefits of secondary schooling[34], which has been shown to be cost-effective elsewhere[35].

This study has some limitations. First, we did not have data on biomarkers for hemoglobin concentration for younger adolescents (aged 10–14 years). Late adolescence, however, is a pivotal period of development when human capital investments can have large effects across the life course[20]. Second, our results are based on WHO hemoglobin cutoffs from a capillary sample to define anemia severity but may differ when using alternative cutoffs[36] or methods to ascertain anemia[37–39]. As a supplementary analysis, however, we used hemoglobin (continuous in g/dl) instead of binary anemia cutoffs to increase the potential relevance of our findings. Third, our household fixed effects models take into account all observed and unobserved characteristics at (or above) the household level but lack control for within-household confounders. Our household fixed effects models assume that unobserved confounders are reasonably similar for adolescents aged 15–18 years within the household, controlling for a wide range of observed demographic characteristics (such as age and birth cohort) and period (survey year) effects. Fourth, as with most cross-sectional analyses, our models are vulnerable to reverse causality. Iron and folic acid (IFA) supplementation, for example, may have been distributed at schools and affect the anemia status of adolescents. The uptake of IFA supplementation programs, however, remains low. In rural West Bengal, less than 10% of adolescents reported taking four tablets offered by the Weekly Iron and Folic Acid Supplementation Programme during the last month[40]; and, in Bihar and Uttar Pradesh, less than 5% of adolescents received IFA supplements in the past year[41]. Our main results were also robust to several sensitivity analyses for the timing of our exposure and outcome, including when using the subset of households where the time between the survey and dropping out of school is short (Supplementary Methods 2). Fifth, our household fixed-effects estimates apply to households with more than one adolescent and may therefore not generalize to households with one adolescent. The sample of households with more than one adolescent, however, was generally similar to the full sample in terms of background characteristics (Table 1). Sixth, our household fixed-effects estimates for the relationship between severe anemia or worse and school attendance are crude, with large CIs which contain zero difference. Given the considerable uncertainty surrounding these estimates for severe or life-threatening anemia, caution should be exercised in making population-level inferences for higher levels of anemia severity.

In summary, based on a large national dataset with measured biomarkers for hemoglobin concentration, we find that having any anemia had few detectable effects on school attendance among adolescents in India. Interventions that target adolescents at risk of anemia may not be sufficient to considerably improve school attendance at the population level in this low-resource setting.

## Data availability

We used unrestricted data which are publicly available upon request from the Demographic and Health Surveys Program (https://dhsprogram.com/). Dataset requests must include contact information, a research project title, and a description of the proposed analysis of the data.

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

## Acknowledgements

We thank study participants in the NFHS surveys and staff at the International Institute for Population Sciences. J.W.D.N. was supported by the Alexander von Humboldt Foundation through a Humboldt Research Fellowship, funded by the German Federal Ministry of Education and Research (grant number: not applicable). O.K. was supported by the Swedish Research Council (grant number: 2019-06396). R.K.R. was supported by the West Bengal State Department of Health and Family Welfare, India (grant number: 114-P&B/HFW-27011/114/2019-NHM SEC). For the publication fee, we acknowledge financial support by Heidelberg University. The funders had no role in study design, data collection and analysis, decision to publish, or preparation of the manuscript.

## Author contributions

J.W.D.N. and S.V. conceived and designed the study. J.W.D.N. conducted the analyses under the guidance of O.K., R.K.R., S.K. and S.V. J.W.D.N. wrote the first draft of the report. J.W.D.N., O.K., R.K.R., S.K. and S.V. reviewed and contributed important revisions to the report. J.W.D.N., O.K., R.K.R., S.K. and S.V. approved the final submitted version of the report. The corresponding author attests that all listed authors meet authorship criteria and that no others meeting the criteria have been omitted. J.W.D.N. and S.V. had full access to all the data in the study and had final responsibility for the decision to submit for publication.

## Funding

## Competing interests

The authors declare no competing interests.
