## [Peer Review File · Communications Medicine]

Reviewers' comments:

Reviewer #1 (Remarks to the Author):

The Major claims made by the authors, do not seem to have acceptable footing.

Line 347 - states that the relationship between severe anaemia and school attendance is pronounced, what does 'pronounced' mean, it is not a technical term in statistics and therefore is not scientifically acceptable.

Line 348 clearly mentions that life-threatening anamiea is NOT significantly associated with low school attendance, however elsewhere this seems to be the main focus/ conclusion.

Line 389 - "Young men who are anemic generally tend to have worse schooling outcomes compared to young women who are anemic." Have you assessed young men who were NOT anaemic, above finding may have nothing to do with anaemia and may be influenced by gender and entering the workforce

Line 418: "Adolescents with moderate anemia, for instance, were less likely to attend school if they were living in regions with lower mean school attendance. "

again have you compared adolescents who were not anaemic? this may have nothing to do with anaemia and everything to do with the fact that the mean school attendance was lower in these areas. and if being anaemic was the reason for them to have lower school attendance, then was this not observed more significantly in those with severe and life-threatening anaemia.

Reviewer #2 (Remarks to the Author):

In this research paper, the authors have undertaken an investigation into the correlation between anemia and school attendance. Their study delves into the nuances of this relationship by considering specific risk factors. The key findings of the study encompass the following:

The research demonstrates that previous observational studies tend to overstate the connection between anemia and academic performance due to factors that have not been accounted for in prior analyses. Notably, adolescents suffering from severe or life-threatening anemia appear to be at a

heightened risk of experiencing adverse educational outcomes. This trend is consistent across various statistical models. Although the prevalence of severe or life-threatening anemia is relatively low in India, the absolute number of affected individuals is substantial. This poses a significant challenge to human capital development in the region.

The paper's novelty, along with its noteworthy findings, will likely garner considerable interest. The statistical analysis is robust; however, there are a few areas of improvement to be addressed:

Given the extensive battery of statistical tests performed, the authors must address the issue of multiple hypothesis testing. Utilizing STATA modules, such as the Romano-Wolf correction, is recommended to adjust p-values. Corrected p-values should be reported exclusively to mitigate the risk of inflated false positives.

To enhance the paper's population-level insights, the authors should consider two approaches: firstly, estimating the proportion of the educational outcome that could be improved by curing severe anemia cases across the entire population, and secondly, determining the total number of adolescents afflicted with severe or life-threatening anemia in the studied population.

The stability of the household fixed effects estimates is an area of concern. The authors primarily relied on a smaller subset of households with more than one adolescent for these estimates. It is imperative to clarify which specific cases are represented in this subset, such as twins or siblings with short birth spacing. Understanding the potential biases introduced by analyzing these cases is critical for the generalizability of the results.

The authors should explore the inclusion of a model that matches children based on birth year, sex, wealth quintile, and district (or PSU). Such models may yield larger sample sizes for estimation than household fixed effects models and can offer additional insights into the relationship between anemia and school attendance.

Reviewer #1:

The Major claims made by the authors, do not seem to have acceptable footing.

We thank the Reviewer for his/her comments, which have further strengthened the paper. The main take-away is that prior observational evidence on the relationship between anemia and school attendance is likely confounded by household-level factors which are not taken into account (Balarajan et al., 2011; Samson et al., 2022). To our knowledge, no other study has conducted a within-household analysis which compares school attendance of adolescents with and without anemia in India. While in more ‘conventional’ (between-household) analyses, anemia is negatively associated with school attendance, in household fixed-effects models, we find little evidence of an association between anemia and school attendance.

Our main result for school attendance is also similar across anemia severity groups. As noted by the Reviewer below, even severe anemia or worse among adolescents, for example, has few detectable effects on school attendance in household fixed effects models (Table 3). Our main result is also robust to alternative definitions of our exposure, such as when using hemoglobin continuously instead of a binary indicator for having anemia (Table S13); alternative covariates; using sample weights; including households with respondents aged 15-24 years old as opposed to aged 15-18 years old (Table S14); and when analyzing each NFHS survey separately (Tables S15 – S17). We also conducted several sensitivity analyses to mitigate potential bias resulting from the timing of the exposure and outcome and found similar results (Text S2).

We have now further clarified our main result throughout the paper. We have clarified that our main exposure is having any anemia (defined as hemoglobin below 11.9 g/dL among non-pregnant women and below 12.9 g/dL among men) and that our main outcome is school attendance. Additionally, we have moved other schooling outcomes which test the sensitivity of our main result to alternative specifications of the outcome to Table S18 in the Appendix (such as literacy, knowledge, and educational attainment). We would be happy, however, to provide additional clarifications and analyses at the behest of the Reviewer and/or Editors.

“Our primary outcome was a binary indicator for an adolescent’s school attendance.”
(p.6, revised manuscript)

“Our exposure was having any anemia defined as hemoglobin below 11.9 g/dL among non-pregnant women and hemoglobin below 12.9 g/dL among men.” (p.7, revised manuscript)

“Table S14. Sensitivity analyses: Household fixed-effects regression results for the relationship between having any anemia and school attendance using alternative

specifications of covariates, interviewer fixed effects, sampling weights, and sample definitions” (p.22, supplementary webappendix)

“Tables S15-17. Sensitivity analysis: OLS regression results for the relationship between anemia severity groups and current school attendance separately by survey year (p.23-25, supplementary webappendix)

“Table S18. Sensitivity analysis: OLS regression results for the relationship between having any anemia and selected learning outcomes available in the NFHS 2005-2019” (p.26, supplementary webappendix)

References:

Balarajan Y, Ramakrishnan U, Özaltın E, Shankar AH, Subramanian SV (2011). Anaemia in low-income and middle-income countries. *The Lancet* 378, 2123-2135.

Samson K et al. (2022). Iron status, anemia, and iron interventions and their associations with cognitive and academic performance in adolescents: a systematic review. *Nutrients* 14, 224.

Line 347 - states that the relationship between severe anaemia and school attendance is pronounced, what does 'pronounced' mean, it is not a technical term in statistics and therefore is not scientifically acceptable.

We agree with the Reviewer and have now removed this term from the paper.

Line 348 clearly mentions that life-threatening anaemia is NOT significantly associated with low school attendance, however elsewhere this seems to be the main focus/ conclusion.

We apologize for any confusion. The first result was for the relationship between life-threatening anemia and school attendance (our primary outcome), shown in Table 3. However, we also showed results for sensitivity analyses where we assessed the relationship between life-threatening anemia and other schooling outcomes (such as literacy, knowledge, and educational attainment). The key idea behind showing results for these additional schooling outcomes was that adolescents with anemia may still attend school but they may perform worse on learning outcomes. Nevertheless, we feel that presenting several schooling outcomes in the main text is

potentially confusing. We have therefore further clarified that our main outcome is school attendance and moved results for other schooling outcomes to the Appendix (Table S18).

“Table 3. Household fixed-effects regression results for the relationship between anemia severity category and school attendance by gender among adolescents aged 15-18 years in India, 2005-2021.” (p.17, revised manuscript)

“Table S18. Sensitivity analysis: OLS regression results for the relationship between having any anemia and selected learning outcomes available in the NFHS 2005-2019” (p.26, supplementary webappendix)

Line 389 - "Young men who are anemic generally tend to have worse schooling outcomes compared to young women who are anemic." Have you assessed young men who were NOT anaemic, above finding may have nothing to do with anaemia and may be influenced by gender and entering the workforce

Yes, the Reviewer is correct. We have assessed young men who were not anemic in two ways. First, in response to the Reviewer’s comment, we have added results for the relationship between any anemia and school attendance when stratifying our sample by gender. This analysis allows the reader to assess changes in school attendance more directly for young men with any anemia vis-à-vis young men without anemia. In these analyses, we find little evidence that school attendance differs between young men with any anemia and without anemia in household fixed-effects models (Table S8), similar to our main results using the pooled sample (Table 2).

Second, to determine whether differences by gender were statistically significant, we also interacted anemia with a male dummy (Table 3). The reader could then add (i) the coefficient on anemia and (ii) the coefficient on the anemia*male interaction to compare school attendance of young men with anemia vis-à-vis young men without anemia. For example, in column 2 of Table 3, the reader could add the coefficients -0.032 (mild anemia) and -0.022 (mild anemia*male interaction) to obtain -5.4 percentage points, which reflects the difference in school attendance between young men with mild anemia vis-à-vis young men without anemia.

“Table S8. Heterogeneity: OLS regression results for the relationship between anemia severity categories and current school attendance stratified by gender (male)” (p.16, supplementary webappendix)

“Table 3. Household fixed-effects regression results for the relationship between anemia severity category and school attendance by gender among adolescents aged 15-18 years in India, 2005-2021.” (p.17, revised manuscript)

Line 418: "Adolescents with moderate anemia, for instance, were less likely to attend school if they were living in regions with lower mean school attendance. " again have you compared adolescents who were not anaemic? this may have nothing to do with anaemia and everything to do with the fact that the mean school attendance was lower in these areas. and if being anaemic was the reason for them to have lower school attendance, then was this not observed more significantly in those with severe and life-threatening anaemia.

Yes, we have now added results when stratifying our models by region to assess changes in school attendance more directly for adolescents with any anemia vis-à-vis adolescents without anemia (Table S12). In these analyses, where the predictor is having any anemia, we find little evidence that school attendance differs between adolescents with and without anemia across regions. In states with low state-level mean school attendance, for example, having any anemia was not significantly associated with school attendance in household fixed-effects models.

“Table S12. Heterogeneity: OLS regression results for the relationship between having any anemia and current school attendance by state-level school attendance” (p.20, supplementary webappendix)

Reviewer #2:

In this research paper, the authors have undertaken an investigation into the correlation between anemia and school attendance. Their study delves into the nuances of this relationship by considering specific risk factors. The key findings of the study encompass the following:

The research demonstrates that previous observational studies tend to overstate the connection between anemia and academic performance due to factors that have not been accounted for in prior analyses. Notably, adolescents suffering from severe or life-threatening anemia appear to be at a heightened risk of experiencing adverse educational outcomes. This trend is consistent across various statistical models. Although the prevalence of severe or life-threatening anemia is relatively low in India, the absolute number of affected individuals is substantial. This poses a significant challenge to human capital development in the region.

The paper's novelty, along with its noteworthy findings, will likely garner considerable interest.

We thank the Reviewer for these comments.

The statistical analysis is robust; however, there are a few areas of improvement to be addressed: Given the extensive battery of statistical tests performed, the authors must address the issue of multiple hypothesis testing. Utilizing STATA modules, such as the Romano-Wolf correction, is recommended to adjust p-values. Corrected p-values should be reported exclusively to mitigate the risk of inflated false positives.

We thank the Reviewer for these comments. Statistical significance has been subject to some controversy and correcting for multiple testing may not correct some of the issues raised elsewhere (Nature Editors 2019). We therefore mostly focused on standard errors (95% CIs) for several comparisons and allowed readers to decide whether the comparisons we show are qualitatively meaningful rather than statistically significant (Gelman et al., 2012). We have now also emphasized in the manuscript that our main outcome is school attendance (Tables 2-3) and moved sensitivity analyses using additional outcomes to the Appendix (Table S18).

As recommended by the Reviewer, we now also provide corrections for multiple hypothesis testing for the relationship between having any anemia and school attendance and between the different anemia severity categories and school attendance. To do so, we used the Romano-Wolf correction, which is the recommended approach by the Reviewer (Clarke et al, 2020). We illustrate corrected p-values and uncorrected p-values in Table R1 below. Our main result for school attendance is consistent after correcting for multiple hypothesis testing. We do not show corrected p-values for sensitivity analyses, however, because these are already a “test” of our main result and we prefer not to adjust our results a second time in these analyses.

References:

Nature Editors (2019). It's time to talk about ditching statistical significance. *Nature*. Mar;567(7748):283.

Gelman A, Hill J, Yajima M (2012) Why We (Usually) Don't Have to Worry About Multiple Comparisons, *Journal of Research on Educational Effectiveness*, 5:2, 189-211.

Clarke D, Romano JP, Wolf M (2020). The Romano-Wolf Multiple Hypothesis Correction in Stata, *Stata Journal* 20(4): 812-843.

Table R1. P-values corrected for multiple hypothesis testing.

Dependent variable: Currently attending school	Survey year indicator		'Conventional' covariates		Household fixed effects		Fixed effects + covariates	
	Model p-value	Romano-Wolf p-value	Model p-value	Romano-Wolf p-value	Model p-value	Romano-Wolf p-value	Model p-value	Romano-Wolf p-value
Predictor	Any anemia (Panel A)							
Any anemia (1=yes)	<0.0001	0.0099	<0.0001	0.0099	0.6297	1.0000	0.4114	1.0000
Predictor	Anemia severity categories (Panel B)							
No anemia	Ref.	Ref.	Ref.	Ref.	Ref.	Ref.	Ref.	Ref.
Mild anemia	<0.0001	0.0099	<0.0001	0.0099	0.0841	0.9604	0.5943	1.0000
Moderate anemia	<0.0001	0.0099	<0.0001	0.0099	0.3189	1.0000	0.4008	1.0000
Severe anemia	<0.0001	0.0099	<0.0001	0.0099	0.2745	1.0000	0.8534	1.0000
Life-threatening anemia	<0.0001	0.0099	<0.0001	0.0099	0.0594	0.9604	0.2751	1.0000

Table R1 shows unadjusted and adjusted p-values for multiple hypothesis testing for the results shown in Tables 2 and 3 in the main text. The sample includes survey respondents who were between 15 and 18 years old at the time of the survey and had a valid hemoglobin test result. P-value adjustments which exceeded one are not valid probabilities and truncated at 1. Source: India NFHS-3, NFHS-4, and NFHS-5 (N=251,401).

To enhance the paper's population-level insights, the authors should consider two approaches: firstly, estimating the proportion of the educational outcome that could be improved by curing severe anemia cases across the entire population, and secondly, determining the total number of adolescents afflicted with severe or life-threatening anemia in the studied population.

We thank the Reviewer for this insightful suggestion. Our household fixed-effects estimates for the relationship between severe anemia or worse and school attendance, however, are relatively crude with large confidence intervals (shown in Table 3 in the main text as well as in Tables S5-S8 in the Appendix). Given the large uncertainty around these estimates for severe or life-threatening anemia, we prefer not to estimate the proportion of the schooling outcomes that could be improved by curing severe anemia cases. Future studies among adolescents suffering from severe anemia or worse could more precisely estimate these proportions.

“Table 3. Household fixed-effects regression results for the relationship between anemia severity category and school attendance by gender among adolescents aged 15-18 years in India, 2005-2021.” (p.17, revised manuscript)

“Table S6. Heterogeneity: OLS regression results for the relationship between life-threatening anemia and current school attendance using a separate binary predictor.” (p.14, supplementary webappendix)

The stability of the household fixed effects estimates is an area of concern. The authors primarily relied on a smaller subset of households with more than one adolescent for these estimates. It is imperative to clarify which specific cases are represented in this subset, such as twins or siblings with short birth spacing. Understanding the potential biases introduced by analyzing these cases is critical for the generalizability of the results.

Our household fixed-effects estimates are stable across a wide range of alternative specifications (e.g., Tables S13-S18). Nevertheless, we agree with the Reviewer that our results may not generalize to households with one adolescent, which we have now addressed in several ways:

First, as recommended by the Reviewer, we have now provided additional descriptions of the adolescents which are included in our sample of households with more than one adolescent. To do so, we assessed the type of relationship of the adolescent to the household head. The most common relationships were son/daughter (80.0%), grandchild (10.0%), and son/daughter-in-law (2.9%) (Table S9 shows the full distribution of types of relationships in the sample used in household fixed effects models). Twinning rates in India are also among the lowest in the world (<9/1,000 births) suggesting that the vast majority of adolescents in our within-household

analyses are siblings. One reason for the relatively low twinning rates in India is that the influence of fertility treatments is still relatively low (Smits & Monden 2011).

Second, we have now provided a supplementary analysis where we included respondents aged 15 to 24 years (as opposed to aged 15 to 18 years) to obtain a subset of households with larger birth spacing between siblings in each household and find similar results (shown in column 6, Table S14). Third, we have now provided more discussion around the generalizability of our findings, including in the Discussion section of the manuscript. Specifically, we find that the sample of households with more than one adolescent is generally similar in terms of background characteristics compared to the full sample (Table 1). Our main results are also consistent when stratifying our sample by the type of relationship to the household head (Table S10).

“Second, we included respondents aged 15 to 24 years (as opposed to aged 15 to 18 years) to obtain a subset of households with larger birth spacing between siblings in each household.” (p.10, revised manuscript).

“Sixth, our household fixed-effects estimates are restricted to households with more than one adolescent and may therefore not generalize to households with one adolescent. The sample of households with more than one adolescent, however, was generally similar to the full sample in terms of background characteristics (Table 1).” (p.21, revised manuscript).

“Table S9. Distribution in the type of relationship of the adolescent to the household head among adolescents living in households with more than one adolescent.” (p.17, supplementary webappendix)

“Table S10. Heterogeneity: OLS regression results for the relationship between having any anemia and current school attendance by relationship to household head” (p.18, supplementary webappendix)

“Table S14. Sensitivity analyses: Household fixed-effects regression results for the relationship between having any anemia and school attendance using alternative specifications of covariates, interviewer fixed effects, sampling weights, and sample definitions” (p.22, supplementary webappendix)

Reference:

Smits J, Monden C (2011). Twinning across the Developing World. *PLoS One*. 2011;6(9):e25239. doi: 10.1371/journal.pone.0025239.

“Table S14. Sensitivity analyses: Household fixed-effects regression results for the relationship between having any anemia and school attendance using alternative specifications of covariates, interviewer fixed effects, sampling weights, and sample definitions”

Dependent variable (DV): Currently attending school (1=yes, 0=no)	(1) Adding age squared, age cubed	(2) Additional covariates	(3) Interviewer fixed effects	(4) Using sample weights	(5) Subsample with recent dropouts	(6) Larger sample (ages 15 - 24 years)
Predictor						
Any anemia (yes=1, no=0)	-0.004 (0.004)	-0.002 (0.004)	-0.002 (0.005)	0.004 (0.006)	-0.002 (0.004)	-0.001 (0.002)
Additional covariates						
Survey year	✓	✓	✓	✓	✓	✓
Indicator for female	✓	✓	✓	✓	✓	✓
Age (years)	✓	✓	✓	✓	✓	✓
Age squared, age cubed	✓	-	-	-	-	-
Month of birth (months)	-	✓	-	-	-	-
Hindu religion (1=yes)	-	✓	-	-	-	-
Indicators for caste/tribe	-	✓	-	-	-	-
Interviewer fixed effects	-	-	✓	-	-	-
Household fixed effects	✓	✓	✓	✓	✓	✓
Total number of households	217,643	202,314	202,314	217,643	185,969	435,492
N , all households	251,401	231,044	231,044	251,401	211,564	574,758
N , households with >1 adolescent	65,459	58,301	58,301	65,459	54,525	191,497
Probability DV, non-anemic	0.683	0.681	0.680	0.668	0.810	0.381
R-squared	0.128	0.133	0.175	0.129	0.091	0.355

Notes: All regressions are ordinary least squares (OLS) models. The dependent variable was a binary indicator for attending school at the time of the survey. In Models 1-4, the sample includes survey respondents who were between 15 and 18 years old at the time of the survey and had a valid hemoglobin test result. In Model 5, the following out-of-school adolescents were excluded from the analysis: age 15 years old with fewer than six total years of schooling; age 16 years old with fewer than seven years of schooling; age 17 years old with fewer than eight years of schooling; and age 18 years with fewer than nine years of schooling at the time of the survey. In Model 6, the sample includes survey respondents who were between 15 and 24 years old (as opposed to 15 and 18 years old) at the time of the survey and had a valid hemoglobin test result. No weights were used except in Model 4. Standard errors in parentheses. *** p<0.01, ** p<0.05, * p<0.1. Source: India NFHS-3, NFHS-4, and NFHS-5 (N=251,401).

The authors should explore the inclusion of a model that matches children based on birth year, sex, wealth quintile, and district (or PSU). Such models may yield larger sample sizes for estimation than household fixed effects models and can offer additional insights into the relationship between anemia and school attendance.

We thank the Reviewer for this suggestion. We explored a model which matches adolescents. However, this approach yields similar results to Model 3 in the manuscript, where we added controls for survey year, age, sex, indicators for household wealth quintile, and area of residence (coefficient on any anemia: -2.5 percentage points; 95% CI: 2.1 – 2.8; shown in Table 2). When matching adolescents on similar characteristics, we find that anemia is associated with a comparable reduction in school attendance (coefficient: -2.0 percentage points; 95% CI: 1.4 – 2.6). Both of these estimates highlight the need to consider all observed and unobserved household-level factors in the relationship between anemia and school attendance and we therefore opted to show the slightly more parsimonious model (Tables 2-3). We would be happy to provide additional analyses, however, at the behest of the Reviewer and/or Editors.

“Table 2. Main OLS regression results for the relationship between any anemia and current school attendance among adolescents aged 15-18 years in India, 2005-2021.” (p. 15, revised manuscript)

“Table 3. Household fixed-effects regression results for the relationship between anemia severity category and school attendance by gender among adolescents aged 15-18 years in India, 2005-2021.” (p.17, revised manuscript)

Reviewers' comments:

Reviewer #1 (Remarks to the Author):

Dear authors

Congratulations on addressing the comments, and verifying the questions brought up.

All the best

Reviewer #2 (Remarks to the Author):

I appreciate the authors for incorporating my suggestions. However, I find their responses somewhat unsatisfactory.

1. Multiple hypothesis testing: The authors assert their reliance on SEs for inference, yet stars indicating statistical significance based on p-values are provided in the tables. To enhance clarity, I recommend either omitting the stars (to redirect the reader's focus away from p-values) or using stars exclusively when the exposure was significant in the R-W correction at 5%.

Is Table R1 displaying the corrected p-values from Table 3? If so, the corrected p-values do influence the conclusions drawn.

2. I find the response here perplexing. Are the authors indicating a lack of confidence in their own estimates (attributed to large CIs), thereby refraining from making population-level inferences? If so, this should be explicitly stated in the limitations, such as:

"Our household fixed-effects estimates for the relationship between severe anemia or worse and school attendance are crude, with large confidence intervals. Given the considerable uncertainty surrounding these estimates for severe or life-threatening anemia, caution should be exercised in making population-level inferences."

Alternatively, if the authors choose not to provide population-level estimates due to the aforementioned reasons, they should at least present a range of estimates for absolute population numbers to convey the magnitude of the issue to readers.

3. Notably, 80% of the cases in the household fixed-effects analysis involve siblings. Assuming that the DHS does not provide preceding birth intervals for adolescents in the sample, I express concern that short birth spacing in these samples might confound results. I kindly request the authors to present their key findings in sub-samples categorized by age differences between siblings (<1 year, <2 years, <3 years, >=3 years).

4. The response is accepted without further comment.

Reviewer #1:

Dear authors, Congratulations on addressing the comments, and verifying the questions brought up. All the best

We thank the Reviewer for these comments.

Reviewer #2:

I appreciate the authors for incorporating my suggestions. However, I find their responses somewhat unsatisfactory. 1. Multiple hypothesis testing: The authors assert their reliance on SEs for inference, yet stars indicating statistical significance based on p-values are provided in the tables. To enhance clarity, I recommend either omitting the stars (to redirect the reader's focus away from p-values) or using stars exclusively when the exposure was significant in the R-W correction at 5%. Is Table R1 displaying the corrected p-values from Table 3? If so, the corrected p-values do influence the conclusions drawn.

We thank the Reviewer for these comments. As recommended, we have now removed all stars from the revised manuscript to redirect the reader's focus away from p-values.

2. I find the response here perplexing. Are the authors indicating a lack of confidence in their own estimates (attributed to large CIs), thereby refraining from making population-level inferences? If so, this should be explicitly stated in the limitations, such as: "Our household fixed-effects estimates for the relationship between severe anemia or worse and school attendance are crude, with large confidence intervals. Given the considerable uncertainty surrounding these estimates for severe or life-threatening anemia, caution should be exercised in making population-level inferences." Alternatively, if the authors choose not to provide population-level estimates due to the aforementioned reasons, they should at least present a range of estimates for absolute population numbers to convey the magnitude of the issue to readers.

The main result of the paper is that prior observational evidence on the relationship between anemia and school attendance is likely confounded by household-level factors which are not taken into account. While in more conventional (between-household) analyses, anemia is

negatively associated with school attendance, we find little evidence of an association between anemia and school attendance in household fixed-effects models (**Table 2**). Moreover, given the large sample of adolescents in our study, we can estimate our main result for the relationship between having any anemia and school attendance extremely precisely.

As a supplementary analysis, however, we also assessed heterogeneity by anemia severity group, including severe anemia or worse. These results are similar with qualitatively small and non-significant coefficients. For severe anemia or worse, however, the number of cases of severe and life-threatening anemia are rather small so that the confidence intervals are considerably larger compared to our main result for any anemia. While we can estimate our main result for any anemia precisely (**Table 2**), as well as for mild and moderate anemia (**Table 3**), our estimates for severe and life-threatening anemia are cruder. Therefore, we still highlight our main result and reject the hypothesis suggesting a link between any anemia and school attendance while we consider the specific supplementary evidence on severe anemia unreliable due to wide CIs (despite a large point estimate) which contain zero difference. As recommended by the Reviewer, we have now added a statement to that effect in the limitations section:

“Sixth, our household fixed-effects estimates for the relationship between severe anemia or worse and school attendance are crude, with large CIs which contain zero difference. Given the considerable uncertainty surrounding these estimates for severe or life-threatening anemia, caution should be exercised in making population-level inferences. Although adolescents with severe or life-threatening anemia represented just 1.6% and 0.6% of our analytical sample, respectively, these numbers represent a large population in absolute terms in India (about 4 and 1.5 million adolescents, respectively, given a population of 250 million adolescents)¹⁹.” (p.16, revised manuscript)

3. Notably, 80% of the cases in the household fixed-effects analysis involve siblings. Assuming that the DHS does not provide preceding birth intervals for adolescents in the sample, I express concern that short birth spacing in these samples might confound results. I kindly request the authors to present their key findings in sub-samples categorized by age differences between siblings (<1 year, <2 years, <3 years, >=3 years).

The Reviewer writes “short birth spacing in these samples might confound results”. However, we controlled for age in our regression models, so that we implicitly also controlled for birth spacing between adolescents (shown in Models 2, 3 and 5, **Table 2**). In sensitivity analyses, we also used alternative specifications of age and included more flexible terms for age (such as age squared and cubed) (**Table S14**). We also used a larger sample of respondents including ages 15-24 years (i.e., with larger birth spacing) as opposed to ages 15-18 years and find similar results. We also considered conditioning on sibling status but these analyses risk inducing additional

biases since it disregards other types of relationships to the household head (such as nieces/nephews and adopted children). Rather than restricting our sample to siblings and stratifying by age, we therefore preferred to add age as a control variable in our models.

Nevertheless, as a supplementary analysis, we have now also stratified our analyses by age differences between siblings, which is the recommended strategy by the Reviewer. To do so, we limited the sample to siblings (children of the household head), which represented about 80% of the sample (**Table S9**). We then stratified this subsample by the difference in age between the oldest and youngest sibling in the household (≤ 1 year, 2 years, 3 years). Consistent with our main result, we find little evidence for a relationship between any anemia and school attendance in household fixed effects models across these groups (shown in **Table R2**). Taken together, these analyses generate additional confidence in the robustness of our main result. We would be happy to provide additional analyses at the behest of the Editors and/or Reviewers.

“In Model 5, we added pre-determined socio-demographic covariates (adolescent’s age and gender) in addition to the household fixed effects included in Model 4” (p. 9, revised manuscript)

“Table 2. Main OLS regression results for the relationship between having any anemia and current school attendance among adolescents aged 15-18 years in India, 2005-2021.”

“Table S14. Sensitivity analyses: Household fixed-effects regression results for the relationship between having any anemia and school attendance using alternative specifications of covariates, interviewer fixed effects, sampling weights, and sample definitions”

Table R2. Sensitivity analyses: Household fixed-effects regression results for the relationship between any anemia and school attendance stratified by birth spacing among siblings

	(1)	(2)	(3)	(4)
Dependent variable (DV): Currently attending school (1=yes, 0=no)	All siblings (children of household head)	Age difference between siblings ≤1 year	Age difference between siblings 2 years	Age difference between siblings 3 years
Predictor				
Any anemia (yes=1, no=0)	-0.003 (0.005)	-0.001 (0.009)	-0.001 (0.006)	-0.009 (0.011)
Additional covariates				
Survey year	✓	✓	✓	✓
Age (years)	✓	✓	✓	✓
Indicator for female	✓	✓	✓	✓
Household fixed effects	✓	✓	✓	✓
Total number of households	24,419	5,622	13,775	5,022
N, households with >1 adolescent	49,808	11,263	27,872	10,673
Probability DV, non-anemic	0.682	0.717	0.668	0.681

Notes: All regressions are ordinary least squares (OLS) models. The dependent variable was a binary indicator for currently attending school at the time of the survey. Standard errors in parentheses. Sample includes survey respondents who were 15-18 years old at the time of the survey, had valid hemoglobin test results, and were living with at least one sibling. Unweighted. Source: India NFHS-3, NFHS-4, and NFHS-5 (N=49,808).

4. *The response is accepted without further comment.*

We thank the Reviewer for these comments.

REVIEWERS' COMMENTS:

Reviewer #2 (Remarks to the Author):

The responses are accepted without further comment. Congratulations to the authors!

Reviewer #2:

The responses are accepted without further comment. Congratulations to the authors!

We thank the Reviewer for these comments.